# High-Vigor Rootstock Exacerbates Herbaceous Notes in *Vitis vinifera* L. cv. Cabernet Sauvignon Berries and Wines Under Humid Climates

**DOI:** 10.3390/foods14152695

**Published:** 2025-07-31

**Authors:** Xiao Han, Haocheng Lu, Xia Wang, Yu Wang, Weikai Chen, Xuanxuan Pei, Fei He, Changqing Duan, Jun Wang

**Affiliations:** 1Center for Viticulture and Enology, College of Food Science and Nutritional Engineering, China Agricultural University, Beijing 100083, China; xiaohan@tust.edu.cn (X.H.); luhc@cau.edu.cn (H.L.); wangyu_0919@cau.edu.cn (Y.W.); xiaobocai55@gmail.com (W.C.); hanxiaoke55@126.com (X.P.); wheyfey@cau.edu.cn (F.H.); chaqduan@cau.edu.cn (C.D.); 2Key Laboratory of Viticulture and Enology, Ministry of Agriculture and Rural Affairs, Beijing 100083, China; 3College of Biotechnology, Tianjin University of Science and Technology, Tianjin 300457, China; xiawang0705@163.com

**Keywords:** aroma, rootstock, wines, LOX-HPL, green flavor

## Abstract

Rootstocks are widely used in viticulture as an agronomic measure to cope with biotic and abiotic stresses. In winegrapes, the aroma is one of the major factors defining the quality of grape berries and wines. In the present work, the grape aroma and wine aroma of Cabernet Sauvignon (CS) grafted on three rootstocks were investigated to inform the selection of rootstocks to utilize. 1103P, 5A, and SO_4_ altered the composition of aromatic volatiles in CS grapes and wines. Among them, 5A and SO_4_ had less effect on green leaf volatiles in the berries and wines, while 1103P increased green leaf volatile concentrations, up-regulating *VvADH2* expression in both vintages. *VvLOXA*, *VvLOXC*, *VvHPL1*, *VvADH1*, *VvADH2*, and *VvAAT* were co-regulated by vintage and rootstock. Orthogonal partial least squares regression analysis (OPLS-DA) showed that the differential compounds in CS/1103P and CS berries were dominated by green leaf volatiles. Furthermore, the concentrations of 1-hexanol in the CS/1103P wines were significantly higher than in the other treatments in the two vintages. 1103P altered the expression of genes in the LOX-HPL pathway and increased the concentration of grape green leaf volatiles such as 1-hexanol and 1-hexanal, while vine vigor also affected green leaf volatile concentrations, the combination of which altered the aromatic composition of the wine and gave it more green flavors.

## 1. Introduction

Grape (*Vitis vinifera* L.) is one of the most important economic crops, cultivated widely across diverse climates [1]. In winegrapes, the aroma is one of the major factors defining the quality of grape berries and wines [2,3]. A complex mixture of volatile compounds—C6/C9 aldehydes/alcohols, higher alcohols, terpenes, C_13_-norisoprenoids, benzene compounds, volatile phenols, and other volatile substances—primarily contribute to a cultivar’s varietal characteristics [3]. These compounds exist in both free and glycosidically bound forms [2]. Free volatiles directly impart fragrance or odor, while the odorless bound forms serve as precursors [4,5]. These precursors represent a potential reservoir of aroma compounds that may be released during winemaking or processing through hydrolysis, subsequently contributing to or altering the overall scent profile.

However, biotic and abiotic stresses often seriously affect the economic performance of grapes in viticulture, especially for winegrapes [6]. Rootstocks are widely used in viticulture as an agronomic measure to cope with biotic and abiotic stresses, which were originally used to address the problem of phylloxera—once devastating to vine growth [7,8]. As research continued, rootstocks were found to affect many phenotypic traits of the scion, such as photosynthesis, vine vigor, yield, berries’ physicochemical indicators (TSS, pH, TA, berry weight), and flavonoid compounds [9,10,11,12]. Recent research has found that rootstocks can be enriched with different microorganisms and that vineyard productivity can be increased by optimizing microbial interactions with rootstocks [13]. These studies generally suggest that different genetic backgrounds of rootstocks confer different indicated characteristics on the scion. High-vigor rootstocks such as 1103P, 110R, and 140R usually give higher yields and stronger growth potential, while low-vigor rootstocks usually result in a more balanced growth potential of the scion. Several studies have been reported on the effect of rootstocks on the berries and wine aroma of scions. 110R, Riparia Gloire, and SO_4_ reduced total ester concentrations in CS berries, whereas 101-14, Ganzin 1, 110R, and 5BB increased C_13_-norisoprenoids concentrations [14]. And Koundouras et al. found that 1103P increased the total aroma substance concentrations in CS berries [9]. Cheng et al. showed that 1103P, Beta, and 5BB could enhance both the total amounts of free-form and bound-form volatile compounds in Chardonnay berries, especially 1103P [15]. Ziegler et al. discovered that BörnerZ can significantly increase the concentration of TDN in Riesling berries [10]. In wines, Vilanova et al. found C_13_-norisoprenoids increased in wines from 110R [16]. 99R and 140Ru led to a higher concentration of total ethyl esters in Merlot wines followed by Gravesac, 110R and 1103P [17].

The climate of the eastern China region is cloudy and rainy during the grape growing season, in such an environment, it is often difficult for the berries to reach the desired ripeness, and the resulting wines are accompanied by a green flavor. Previous studies have shown that these green flavors are frequently caused by the presence of C6/C9 aroma substances, which are also called GLVs, and the lipoxygenase–hydroperoxide lyase (LOX-HPL) pathway is an important pathway for the production of volatile GLVs. LOX, HPL, alcohol dehydrogenases (ADHs), and acyltransferase (AAT) are the major enzymes in this pathway [18,19]. We hypothesized that rootstock and vintage together cause changes in genes in the LOX-HPL pathway and that these changes can change the aroma components of grapes and wines that can be perceived by tasters. As a result, we investigated three graft combinations in eastern China over three vintages to confirm berry quality and winemaking in two consecutive vintages, and hoped to find a rootstock that reduces the aromatic substances of green flavor wine to improve wine quality, which could provide advice in rainy, cloudy regions like eastern China.

## 2. Materials and Methods

### 2.1. Chemicals

Sodium hydroxide, citric acid, sodium chloride, and disodium, citric acid, and disodium phosphate were obtained (Beijing Chemical and Pharmaceutical Factory, Beijing, China). Polyvinylpolypyrrolidone, D-gluconic acid lactone, and C6-C24 n-alkanes and aroma compound standards were obtained (Sigma Aldrich, Burlington, MA, USA).

### 2.2. Experimental Site and Sampling

The experiment was conducted at China Agricultural University’s Shangzhuang experimental station (40°14′ N, 116°20′ E, 49 m altitude) for 3 consecutive vintages (2018–2020). In this field experiment, 3 rootstocks (1103P, SO_4_, 5A) were used to graft Cabernet Sauvignon (CS) clone 685. These vines were spaced at 2.5 × 1.2 m, in north–south rows, and were planted in 2012. Furthermore, a modified vertical shoot positioning training system was employed in the vineyard [15], which retained 12 to 15 nodes per linear meter. Each graft combination contained 3 replicates and each replicate had 15 vines. Nutrition and pest control methods were conducted in accordance with local industry norms. Grape growing season average monthly temperatures, sunlight hours, and precipitation were gathered from the China Meteorological Data Interchange Platform (https://data.cma.cn/, accessed on 15 June 2025) (Appendix A). Soil sampling referred to the previous study by Han et al. [20] and briefly described as follows: 9 sample plots were randomly taken in the vineyard and mixed into 3 replicates, and each sample plot was taken at 5 different depths (0 to 20, 20 to 40 cm, 40 to 60 cm, 60 to 80 cm, and 80 to 100 cm). Soil basic physicochemical indices were tested with specific reference to Brun et al. [21]. The Kjeldahl method was used to determine N in soil, which consisted of 3 steps: sample digestion, distillation, and ammonia determination [22]. The vine growth indexes are shown in Appendix A, and the physicochemical parameters of graft combinations in berries and wines are shown in Appendix A, and the physical-chemical indices of vineyard soil are shown in Appendix A [23,24].

### 2.3. Berry Sampling and Winemaking

At the annual harvest, 300 grape berries were taken from each biological replicate for the determination of physicochemical indicators and aroma compounds, respectively. Additionally, in 2019 and 2020, 20 kg of clusters were collected from each biological replicate for winemaking. The clusters are first de-stemmed using a de-stemming machine and then crushed by hand. Then placed in 20 L stainless steel vessels, and 0.6 g of SO_2_ and 0.4 g of pectinase (Optivin, VinCru Pty Ltd., Melbourne, Australia) were added to the must and controlled to reach a total SO_2_ content of approximately 60 mg/L. A pre-fermentation maceration was performed for 24 h at 18–20 °C, and 3.6 g of commercial Lalvin D254 yeast (Laffort, Bordeaux, France) was activated and inoculated into the must. Alcoholic fermentation was carried out in a temperature-controlled winery (23–25 °C). The temperature and specific gravity of the must were measured regularly every day during fermentation, and the skins were punched down twice a day, morning and evening. And the skins and seeds were then removed when the reducing sugar level dropped below 1 g/L, and 0.02 g of lactic acid bacteria (Lalvin31, Lallemand Inc., Saint-Simon, France) was added to start malolactic fermentation. The alcoholic fermentation went on for about 7 days and the malolactic fermentation went on for about 1 month. When malolactic fermentation was completed, 1.2 g of SO_2_ was added to the wine to control the total SO_2_ content to approximately 80 mg/L. Afterward, the wine was bottled in 750 mL bottles and refrigerated in the cellar (12–16 °C, no light) until analysis. After malolactic fermentation, the wines were aged in fermentation tanks for 1 month and then bottled.

### 2.4. Quantification of Aromatic Volatiles by GC–MS/MS

The experimental methods were referred to from Wang et al. [25] as follows: In liquid nitrogen, 60 g of frozen berries without seeds were ground into powder together with polyvinylpolypyrrolidone (0.5 g) and D-gluconic acid lactone (0.5 g). The frozen grape powder was placed in 50 mL centrifuge tubes. The centrifuge tube was immediately capped and stored for 6 h at 4 °C for cold stabilization and equilibration of volatile component extraction. Then, the homogenate was spun at 8000× *g* for 15 min to obtain the supernatant [25]. The free-form volatile chemicals were extracted from clear juice using the headspace solid phase microextraction, and the extraction of bound-form volatile chemicals referred to Wang et al. [25]. The HS-SPME extraction was conducted using a CTC-CombiPAL autosampler (CTC Analysis, Zwingen, Switzerland) that was mounted with a 2 cm SPME fiber coated with DVB/CAR/PDMS (50/30 μm, Supelco, Bellefonte, PA, USA). The aromatic profile of grape and wine were characterized using gas chromatography-mass spectrometry (GC-MS) (Aligent, Santa Clara, CA, USA). Both GC systems were equipped with identical HP-INNOWAX capillary columns (J&W Scientific, Folsom, CA, USA; 60 m length × 0.25 mm internal diameter, 0.25 μm film thickness). Grape aromatic profile was obtained with an Agilent 6890 gas chromatograph and an Agilent 5973 mass spectrometer, and wine aromatic profile was obtained with an Agilent 7890 GC and an Agilent 5975 MS. Identification was performed via mass spectra matching through the standard NIST11 library. The chromatographic and mass spectral data obtained were matched with the mass spectral NIST 11 library, and the chromatographic peaks with a match of ≥80 with the target substances were selected. The retention indices of the peaks were compared with the retention indices (RIs) provided by the NIST library, and individually with the literature values to determine the name of the substance corresponding to each peak. Specific chromatographic information is shown in Appendix A. The retention indices of the peaks were compared with the retention indices (RIs) provided by the NIST library and individually compared with literature values to determine the names of the substances corresponding to each peak. The quantification of aroma compounds was conducted based on previous studies [26,27,28]. Briefly, the calibration curves for aroma standards were established based on the mixed standard solution that was diluted into 15 levels using a synthetic model matrix. Artificial grape juice was made by mixing distilled water with 200 g/L of glucose and 7 g/L of tartaric acid, adjusting the pH to 3.3 using 5 M NaOH. Simulated wine mixtures were made using 12% ethanol and water solutions with 5 g/L tartaric acid, and then the pH was adjusted to 3.8 using the same alkali. Volatile analytes lacking dedicated calibration curves were quantified through regression models derived from structural analogs sharing comparable carbon chain lengths and functional groups. Odor activity values were determined by dividing individual compound concentrations by their respective sensory perception thresholds. Threshold data and associated aroma classifications were sourced from Francis et al. [27].

### 2.5. Sensory Analysis

The wines were subjected to sensory tasting after approximately 3 months of bottle storage in the cellar. The sensory experiments were performed by 12 trained wine panelists (7 females and 5 males) between the ages of 23 and 29 who were chosen from the Center of Viticulture and Enology (CFVE) after receiving sensory training for more than a year. They were trained in the identification of flavors such as dried fruit, fresh fruit, botanical flavors, red berries, black berries, smoky, floral, and honey flavors and have participated in more than 10 formal sensory experiments. Wine body, acidity, bitterness, and astringency were also evaluated. For the tasting, 3 replicate wine samples were first mixed and then tasted in a completely randomized order. Each panelist conducted 3 repetitions of the evaluation for each sample. The tasting was scored mainly on aroma, taste, and overall profile, using a 10-point scale. Prior to participation, all subjects provided written informed consent affirming voluntary involvement in this study, in compliance with the Declaration of Helsinki (1975). Ethical authorization for human subject research was formally issued by the Research Ethics Committee of China Agricultural University (Approval No. CAUR-20220711).

### 2.6. Total RNA Extraction and Real-Time qPCR Assay

Seedless berry samples (about 10 g) were ground in liquid nitrogen with 1 g of polyvinylpolypyrrolidone (PVPP, Macklin, Shanghai, China). From the resulting powder, 0.1 g portions were set aside for RNA isolation, and the rest was thrown away. Total RNA was extracted using the Spectrum™ Plant Total RNA Kit (Sigma-Aldrich, Wuxi, China) following the instructions provided, which included using DNase I to remove any genomic DNA contamination. The quality of the RNA was checked using a 1.5% agarose gel, and its purity and amount were measured with a NanoDrop 2000 spectrophotometer, showing OD260/OD280 ratios between 1.8 and 2.2. First-strand cDNA was made in 20 μL reactions using 1 μg of total RNA and HiScript^®^ II Q RT SuperMix for qPCR + gDNA wiper (Vazyme, Nanjing, China). Transcript analysis of key genes in the LOX–HPL pathway was performed by real-time qPCR using the SYBR green method on a CFX96 Real-Time System, with the ubiquitin gene as the reference. The experimental methods are referred to by Qian et al. [29] as follows: Each reaction (10 μL) contained 5 μL SYBR (Vazyme, Nanjing, China), 4.5 μL ddH_2_O, 1/6 μL cDNA, 1/3 μL primer mixture (forward primer and reverse primer, 10 μM). Reverse-transcription PCR (RT-PCR) was adapted from Tan et al. with some modifications [30]. cDNA templates underwent initial denaturation (95 °C, 30 s) prior to 40 amplification cycles with the following thermocycling parameters: denaturation at 94 °C for 10 s, annealing at 60 °C for 30 s, culminating in a dissociation analysis (60–95 °C ramp). The specific gene primers for the lipoxygenase-hydroperoxide lyase (LOX-HPL) pathway and the reference genes are listed in Appendix A [31,32,33,34]. For berry specimens representing distinct graft combinations, RNA isolation was conducted in triplicate biological replicates, with each extract subjected to triplicate technical replications during real-time quantitative PCR (qPCR) analysis.

### 2.7. Statistical Analyses

The analysis of variance (ANOVA) was conducted with R 4.0.5 and a significance level of *p* < 0.05 (Duncan’s multiple range test). Origin 2021 was used to create the images (OriginLab Corp., Northampton, MA, USA). SMICA 14.1 was used to conduct principal component analysis (PCA) and OPLS-DA (Umetrics, Umea, Sweden).

## 3. Results

### 3.1. Meteorological Conditions and Soil Indices

Analysis of climatic conditions across the three vintages revealed distinct patterns: 2018 was characterized by high temperatures and abundant rainfall, 2019 featured sufficient sunlight with reduced precipitation, while 2020 exhibited lower light levels and the lowest effective cumulative temperature (Appendix A and Figure 1A–C). Soil texture, a key indicator of vineyard soil characteristics, is typically defined by particle size composition. An optimal particle composition facilitates the retention and supply of soil nutrients and moisture. Based on particle size distribution and the International Soil Texture Classification system, the vineyard soil (BJ) was classified as sandy clay loam (Figure 1D). Key soil physicochemical properties and mineral concentrations were also quantified. Cation exchange capacity (CEC), representing the total quantity of exchangeable cations adsorbable by soil colloids, was a critical parameter for assessing soil fertility retention capacity, guiding soil amelioration, and informing rational fertilizer application [35]. Measured CEC values in the vineyard soil profile ranged from 11.7 to 18.03 cmol/kg, indicating a relatively high capacity for nutrient retention. Soil bulk electrical conductivity (EC) was reported to correlate closely with soil moisture content, salinity, and texture [36]. Other soil physicochemical indices are shown in Appendix A.

### 3.2. Effects of Rootstocks on Volatile Compound Profiles in Berries and in Wines

A total of 60 volatile compounds were quantified across three graft combinations and own-rooted CS berries (SupplementalAppendix A). These aroma substances were categorized into the following: C6/C9 compounds, terpenes, higher alcohols, benzenes, acids, esters, C_13_-norisoprenoids, volatile phenolic, and others. To better understand how the aroma compounds in the grape berries relate to the wine, we would look at the total amount of aroma substances in the berries, which included both free-form and bound-form aroma compounds, in the next analysis. As shown in Figure 2A, the C6/C9 compounds were the most abundant in the grapes. 1103P tended to increase the C6/C9 substance concentration and showed a significantly higher level than CS and other graft combinations in 2020. Furthermore, CS/5A and CS/SO_4_ showed no significant difference in C6/C9 substance concentration in 2018 and 2019 compared to CS but significantly reduced it in 2020. Also, 1103P and SO_4_ did not significantly affect the levels of C_13_-norisoprenoids in the berries over the three years, while 5A only lowered the levels of C_13_-norisoprenoids in the berries in 2020 (Figure 2D). 5A and SO_4_ lowered the amounts of terpenes in berries in 2018 and 2020, while 1103P seemed to lower the amounts of terpenoids in 2020. For the concentrations of benzenes and acidic compounds, the different graft combinations did not show differences in 2019, and SO_4_ tended to reduce both types of substances in 2020 (Figure 2E,F). For higher alcohols, different graft combinations did not show differences in 2020, and 5A had a tendency to increase the concentration of higher alcohols in 2019 (Figure 2B). Volatile phenolic compounds, which could contribute to irritating flavors such as smoky, spicy, animal, medicinal, and chemical notes, are less abundant in the berries and primarily consist of guaiacol and phenol (Figure 2G). In this study, the different graft combinations were similar to CS in 2018 and 2019, but in 2020, 1103P and SO_4_ significantly raised the levels of volatile phenolics, while 5A significantly lowered them.

In addition, a total of 75 aroma compounds were detected in the wines of three graft combinations and CS in two vintages (Appendix A). These aroma compounds could also be classified into C6/C9 compounds, terpenoids, higher alcohols, benzenes, acids, esters, C_13_-norisoprenoids, and volatile phenolic compounds. Higher alcohols and esters were the main compounds in wine. Three rootstocks significantly reduced the concentration of higher alcohols in 2020. In this study, SO_4_ was found to significantly reduce the content of higher alcohols in the wines in 2020, which showed a consistent effect of SO_4_ on the content of higher alcohols in the wines (Figure 3B). Interestingly, 1103P and SO_4_ significantly reduced the concentration of esters in wines in 2019, while only showing a trend of reduction in 2020 (Figure 3H). 1103P reduced the concentration of benzene substances in wine, while increasing the concentration of C6/C9 alcohols. 5A and SO_4_ only had a trend to reduce benzene compounds in 2020 (Figure 3G). All three rootstocks significantly reduced the concentration of volatile phenolics in wine (Figure 3F). 5A and 1103P had a tendency to reduce terpenes in wine, while SO_4_ significantly increased terpenes in wine in 2019. In 2019, all three graft combinations had significantly lower terpene concentrations than CS, while in 2020, the CS/SO_4_ combination significantly increased terpenes in the wines (Figure 3C).

### 3.3. PCA and OPLS-DA of Compounds in Berries and in Wine

PCA was employed to delineate aroma compound profiles across grape graft combinations. The analysis revealed no significant segregation among rootstock treatments but demonstrated pronounced interannual variation (Figure 4A), indicating vintage constitutes the predominant differentiating factor for berry volatile composition. Principal component 1 (PC1) accounted for 49.2% of total variance, while PC2 explained 35.1% of the variance, collectively representing 84.3% of dataset variability. It was obvious to distinguish the three vintages, but it was difficult to distinguish between different rootstock combinations, and the vintage played a decisive role in the aroma. To mitigate confounding effects from vintages and rootstock-specific contributions to scion metabolite profiles, OPLS-DA was employed. The model was constructed using 70% of the dataset for training and 30% for validation, with predictive validity further assessed through 200 permutation tests to prevent overfitting. When we tried to fit the model with data from three vintages, the model was over-fitted and unreliable, so we fitted the model with data from two vintages, 2019 and 2020, and obtained a reliable model. As shown in Figure 4B, PC1 first distinguishes between the CS and CS/1103P combinations, and PC2 distinguished between years. Their key compounds included (E)-3-hexen-1-ol, isobutyl octanoate, 1-octen-3-ol, nonanal, ethyl decanoate, (E)-2-octenal, hexanoic acid, phenylethyl alcohol, (Z)-3-hexen-1-ol, benzyl alcohol, 1-hexanol, and (E, Z)-2,6-nonadienal (Figure 4B and Appendix A). Unfortunately, even after fitting the model with two vintages of aroma compounds, we were still unable to obtain a reliable model for comparing CS/SO_4_ to CS and CS/5A to CS. This further indicated that the two graft combinations, CS/SO_4_ and CS/5A, did not differ significantly from the overall aroma profile of CS, while 1103P had a clearer effect on the concentration of aroma substances in the scion. The vintage had a much greater effect than the rootstocks, which could mask the differences caused by rootstocks in the grape aroma. From another perspective, those aroma compounds that were more influenced by vintage are precisely the more plastic compounds, while these more plastic compounds tended to be false-positive marker compounds in single vintage experiments. Therefore, for cultivation experiments in the field, it was necessary to conduct more than two consecutive vintages of experiments to ensure the reliability of the results.

PCA was also utilized in wines to categorize the various graft combinations and CS in two vintages. PC1 separated the two vintages by accounting for 49.3% of the total variance (Figure 5A). Wine in 2020 included more C6/C9 alcohols and volatile phenolics, and wine in 2019 included more terpenoids. In addition, OPLS-DA was used again to search for differential compounds imparted to wines by different rootstock combinations. PC1 differentiates between CS/1103P and CS, and PC2 differentiates between the two vintages; there were 26 different types of differential compounds (VIP > 1) between the CS/1103P combination and CS (Figure 5B and Appendix A). Furthermore, PC1 differentiated between CS/SO_4_ and CS, whereas PC2 differentiated between the two vintages; there were 29 different types of differential compounds (VIP > 1) between the CS/SO_4_ combination and CS (Figure 5C and Appendix A). When it came to the CS/5A combination, the mode was overfitted. Therefore, we performed OPLS-DA on CS/5A and CS wines from 2019 and 2020, respectively, and then used the intersection of the characteristic compounds of these two vintages as the characteristic compounds of CS/5A and CS (Figure 5D,E and Appendix A).

### 3.4. Volatile Compounds and Their OAVs in Wine and Sensory Tasting

OAVs, a quantitative metric for assessing volatile compounds’ contribution to wine aroma profiles [37], were determined for detected volatiles. The OAV of each compound was derived by dividing its experimental concentration by the corresponding sensory perception threshold documented in established literature sources. And OAVs were calculated by dividing the concentrations of volatile compounds in wines by their sensory thresholds. Aroma compounds with OAVs > 1 were generally considered to be clearly perceived by the panelists [38]. In this study, we identified 18 compounds with OAVs > 1, which included 7 types of esters, 3 types of higher alcohol, 2 types of benzene compounds, 2 types of terpene compounds, 2 types of C_13_-norisoprenoids, and 2 types of acid (Appendix A). *β*-damascenone, a volatile molecule with high OAVs, smells like honey, tropical fruit, quince, and apple and may indirectly affect wine fragrance by amplifying the fruity notes of ethyl esters [39]. OAVs for *β*-damascenone did not differ significantly among graft combinations in 2019, but SO_4_ improved OAVs significantly in 2020 (Appendix A). 1-hexanol, as the only C6 alcohol substance with OAVs > 1, contributes herbaceous, grassy, and woody aromas that could be intuitively perceived by wine panelists. The OAVs of 1-hexanol were significantly higher in CS/1103P wines than in CS and other graft combinations over two consecutive vintages, indicating that 1-hexanol is a key substance influencing the greenness of the wine.

It is encouraging to see that CS/1103P had a higher green flavor score in the sensory tasting (Figure 6), which was consistent with our inferred results based on aroma substances with OAVs > 1. CS had the highest overall berry score, but there was a slight difference between the two vintages, with CS having the highest red berry score in 2019 and CS having the highest black berry score and flower flavor in 2020, which might be caused by more esters in CS wines. It has been shown that ethyl propionate, ethyl 2-methylpropionate, and ethyl 2-methylbutyrate give wines more black berry flavors, while ethyl butyrate, ethyl caproate, ethyl caprylate, and ethyl 3-hydroxybutyrate give wines more red berry flavors [40]. While CS/5A had a lighter green flavor, it received the highest overall evaluation score among all samples. Through sensory evaluation, 1103P did add a green flavor to the wine, and the substance that contributed to the green flavor, among the OAVs > 1 was 1-hexanol. Interestingly, OPLS-DA also showed that 1-hexanol was the differential compound between CS/1103P wines and CS wines (VIP > 1), and the ANOVA also showed that CS/1103P wines had significantly higher levels of 1-hexanol than CS wines (Appendix A). Therefore, we could conclude that 1-hexanol was the key GLV that distinguishes CS/1103P wines from CS wines. This difference was also reflected in the grape. A study also indicated that 1-hexanol was considered a secondary source of differences among four grape varieties [41]. In addition, it was found that the influence of 1103P on aroma was always concentrated on the C6/C9 substances in the grapes. Therefore, we analyzed the critical genes on the C6/C9 compounds synthesis pathway to further explain why 1103P contributes to green flavor.

### 3.5. Expression of Genes in LOX-HPL Pathway

C6 alcohol/aldehyde were considered to be characteristic compounds of non-muscat grape varieties such as Cabernet Sauvignon, Monastrell, and Tempranillo [42]. Based on previous studies, it was discovered that the LOX-HPL pathway controls the synthesis of C6/C9 compounds [43], so six key genes (*VvLOXA*, *VvLOXC*, *VvHPL1*, *VvADH1*, *VvADH2* and *VvAAT*) were chosen to perform fluorescence quantitative PCR. In 2019, 1103P significantly up-regulated *VvLOXA*, *VvHPL1*, *VvADH1*, *VvADH2*, and *VvAAT*, while 5A significantly down-regulated *VvADH2* and SO_4_ significantly up-regulated *VvLOXA* (Figure 7A). According to previous reports, we know that the unsaturated fatty acids linoleic and linolenic acids are produced by LOX and could be classified into 9-LOX and 13-LOX. Therefore, we could infer that 1103P first up-regulates *VvLOXA*, that 9-hydroperoxides and 13-hydroperoxides can be generated in the presence of *VvLOXA*, and that *VvHPL1* catalyzes the production of hexanal by 13-hydroperoxides, which is converted to C6 alcohol by *VvADH1* and *VvADH2*, leading to a significant increase in 1-hexanol in berries and wine. C6 esters were produced by the action of *VvAAT*, but the concentration of C6 esters in ripe grape berries was relatively low. In 2020, 1103P significantly up-regulated *VvADH2* while significantly down-regulating *VvAAT*, 5A significantly up-regulated *VvHPL1* and *VvADH1* and SO_4_ significantly up-regulated *VvlOXC* and *VvADH1*. *VvLOXA*, *VvHPL1*, *VvADH1*, and *VvAAT* did not show a consistent pattern compared with 2019, which suggested that vintage was also an important factor affecting gene expression and that there may be an interactive effect between vintage and rootstock. Therefore, we performed a two-way ANOVA on vintage and rootstock and found that vintage had a highly significant effect on *VvLOXA*, *VvHPL1*, *VvADH1*, *VvADH2*, and *VvAAT*. And rootstock had a highly significant effect on *VvHPL1*, *VvADH1*, *VvADH2*, and *VvAAT*, while vintage and rootstock co-regulated *VvLOXA*, *VvLOXC*, *VvHPL1*, *VvADH1*, *VvADH2*, and *VvAAT* (Figure 7C).

Combined with the GLV concentrations in the grape, it was found that 1103P increased the concentrations of GLVs in grape and wine in both vintages (2019 and 2020), but we considered the main reasons for this to be different. The impact of rootstock on scion aroma substances should be considered from two perspectives: On the one hand, rootstock influenced the accumulation of GLVs by regulating the expression of genes on the LOX-HPL pathway, which was the dominant factor in the sunny and dry vintage (2019). Rootstocks were subjected to less environmental stress in this environment, resulting in less interference with genes that regulate the formation of grape aroma quality. On the other hand, excessive vegetative growth limited reproductive growth and delayed the berry ripening process, resulting in the accumulation of more GLVs in the berries, which was the main factor in the rainy and cloudy vintage (2020). In this environment, the rootstock responded more to environmental stress than it did to regulate the formation of berry aroma quality, which interfered with gene expression and led to some gene expressions that were inconsistent with those from 2019. As shown in Appendix A, the pruning weight of the CS/1103P combination in 2020 was higher than both the pruning weight of the other treatments in the same year and the CS/1103P combination in 2019.

In combination with the differential compounds in wine, 1-hexanol was the key compound to distinguish CS/1103P and OAVs > 1, so we concluded that the green flavor in the wine was contributed by 1-hexanol. When we compared 1-hexanol alone, 1-hexanol was significantly higher than the other rootstock combinations in the wines in two consecutive vintages (Figure 8A). But the difference was not significant in the berries. However, when we summed up the content of 1-hexanol and 1-hexanal in the berries, it was found that the content in CS/1103P was consistently higher than in the other rootstocks (Figure 8B–D). In the wines, only 1-hexanol but not hexanal was detected. Therefore, we speculated that hexanal in the berries was converted to 1-hexanol during fermentation, resulting in a significant increase of 1-hexanol in the CS/1103P wine. This also seemed to explain why it was difficult to have a direct correlation between the level of a substance in the grape alone and the level of the corresponding substance in the wine.

## 4. Discussion

Among the C6/C9 substances, (E)-2-hexenal and hexanal were the most abundant aroma compounds [14]. 1103P mainly promoted the accumulation of these two compounds, which led to a significant increase in the total C6/C9 substance concentration (Appendix A). Cheng found that 1103P significantly increased the concentration of C6/C9 compounds in Chardonnay [15]. A recent study also indicated that 1103P has more aromatic compounds, abundant green-vegetable/astringent notes, and more defect-causing compounds [44]. These studies support our findings, and our results further demonstrate that 1103P confers more C6/C9 compounds to grape berries, particularly in vintages with high rainfall. SO_4_ reduced the content of most compounds in grape berries, but 5A had a relatively minor impact on the aroma components of CS grape berries. This might be due to the imbalance between vegetative growth and reproductive growth. The more vigorous vegetative growth of CS/1103P and CS/SO_4_ might have changed the metabolic flow of some aroma compound precursors, thus having a greater impact on the aroma components of CS grape berries, especially in years with more rainfall. The vegetative growth of CS/5A was more consistent with that of own-rooted, so the composition characteristics of its aroma compounds were also more similar.

In wine, we found that 1103P increased the levels of C6/C9 alcohols and acids while decreasing the levels of higher alcohols, benzenes, esters, and volatile phenolic compounds compared to own-rooted. It has been shown that Merlot wine made from the Merlot/SO_4_ combination has a lower higher alcohols content [17]. Among the many compounds, it was found that the concentration levels of compounds in the grape and wine do not consistently correspond to each other, which was mainly caused by the complex winemaking process. Many compounds undergo transformation losses during the fermentation process. When we used OPLS-DA to fit models for grape berry and wine compounds from different rootstock combinations, we found that the fitted model for wine was better than the fitted model for grape berry. This showed that the differences in aroma substances between different graft combinations were amplified by fermentation, and it was clear that we were able to successfully build a reliable fit model when differentiating wines. This was partly due to the fermentation process creating more complex aromas in the wines, and as the number of compound species increases, the dependent variable used to evaluate differences between grafted combinations and CS also increases, making it easier to construct a reliable model. Furthermore, the sensory aroma profile of wine emerged from complex binding phenomena between volatile constituents and matrix macromolecules [45]. Fermentation aromas were more closely related to the grape’s primary metabolites, such as sugars, acids, and amino acids, and differences in the grape’s primary metabolites could also cause changes in the wine’s aroma composition.

Interestingly, among the many complex compounds, we found that the C6/C9 substances in grapes and wine had a better correspondence. Among them, 1103P greatly raised the overall amount of C6/C9 compounds in both grapes and wines, and this difference was also consistent across different years. This variation might be because C6/C9 compounds were very abundant in grapes. Grapes reduced C6/C9 aldehydes to C6/C9 alcohols during the fermentation process, allowing them to enter the wine without complex interference. Although the selection of yeast strains and fermentation conditions would also affect the reduction efficiency, the precursor concentration in the fruit was the most fundamental determining factor.

This study also investigated the effect of rootstock on the expression of genes in the C6/C9 compound metabolic pathway. The effect of rootstock on genes in grapes varied accordingly under different vintage conditions, with vintage and rootstock jointly determining changes in LOX-HPL pathway genes. Although gene expression was influenced by the vintage, we still found that 1103P up-regulated *VvADH2* in both vintages, indicating that this gene was less affected by the vintage and was the key gene affected by 1103P. Qian et al. [29] compared the expression of LOX-HPL pathway genes in four varieties and found that *VvLOXA*, *VvHPL1*, and *VvAAT* were targets that altered the composition of GLVs in grape varieties. A recent study also showed 5BB and 1103P up-regulated five *VvGTs* and nine genes from the LOX and MEP pathways in Marselan berries. 1103P increased the concentrations of C6 alcohols, C6 aldehydes, and citronellol from the Marselan berry [46]. However, in merlot grapes, 1103P significantly down-regulated the *VvLoXA*, *VvADH1*, *VvADH2*, and *VvADH3* expression levels, which indicated that different scion varieties also affect the gene expression level. It has also been studied that seawater irrigation affects the expression of *VvLOXA*, *VvHPL*, *VvADH*, and *VvAAT* in grapes [47]. This shows that genes on the LOX-HPL pathway are highly plastic and susceptible to environmental and cultivation practices. A previous study showed that excessive rainfall caused branch depression, allowing CS/1103P to be pruned more aggressively [48], which affected grape development and flavor compound accumulation. Another study showed that proper water deficiency was beneficial to wine composition, global quality, and sensory perception for high-vigor rootstocks (1103P, 140Ru) [44], which explains why 1103P only has a tendency to increase GLVs in dry and sunny vintage, while significantly increasing GLVs in rainy and cloudy vintage.

Considering the sample size and the focus on volatile compounds, grapevine standing conditions, and vintage of this study, there are some limitations in this study, and there is a need to expand the sample size and vintages of this study to further validate the results of this study in future research.

## 5. Conclusions

The rootstock’s effect on grape quality is first seen in the berries’ aroma compound concentrations, and as vinification continues, the rootstock’s effect on the grape’s aromatic substances is seen in the wine, even though this difference does not correspond well for many substances. GLVs were the most abundant compounds in the berries, but in wine, only 1-hexanol OAVs > 1 can be perceived by the panelists, and most GLVs do not exceed the threshold. We can assume that the green flavor in the final wine is mainly contributed by 1-hexanol. In terms of the aromatic profile of Cabernet Sauvignon wines, the use of vigorous rootstock 1103P increases the risk of green aroma and should therefore not be preferred in humid climates.

## Figures and Tables

**Figure 1 foods-14-02695-f001:**
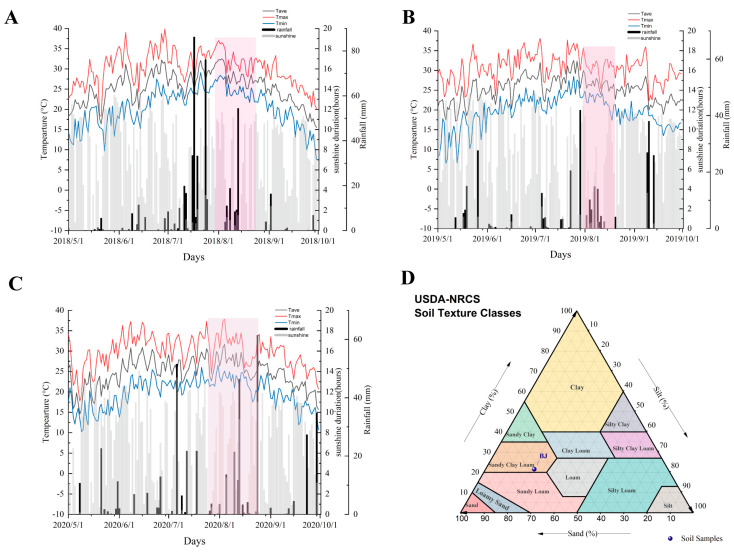
Meteorological data of Beijing in 2018–2020 (**A**–**C**). Texture of vineyard soil (**D**). note: pink sqaure represents the grape berries at veraison; BJ stands for the abbreviation of the experimental site, Beijing.

**Figure 2 foods-14-02695-f002:**
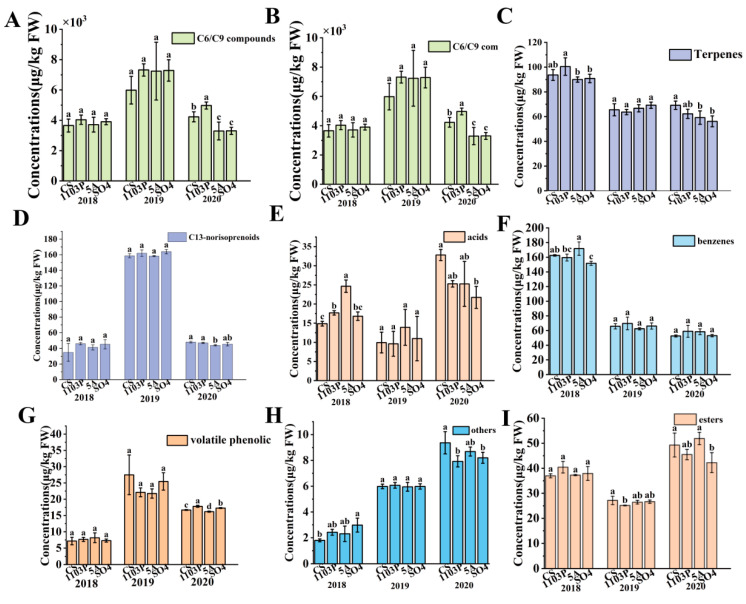
Aroma compound (C6/C9 compounds, higher alcohols, terpenes, C_13_-norisoprenoids, acids, benzenes, volatile phenolic, others, esters) concentrations of different grafted combinations of Cabernet Sauvignon in berries (**A**–**I**). Note: different letters mean significant differences among the four treatments on the basis of Duncan’s multiple range test as *p* < 0.05.

**Figure 3 foods-14-02695-f003:**
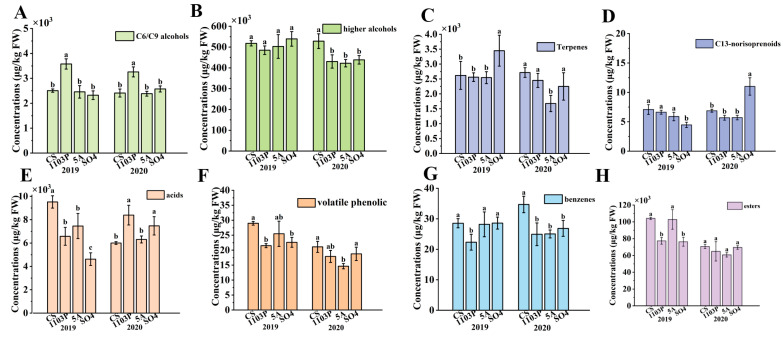
Aroma compound (C6/C9 compounds, higher alcohols, terpenes, C_13_-norisoprenoids, acids, volatile phenolic, benzenes, esters) concentrations of different grafted combinations of Cabernet Sauvignon in wines (**A**–**H**). Note: different letters mean significant differences among the four treatments on the basis of Duncan’s multiple range test as *p* < 0.05.

**Figure 4 foods-14-02695-f004:**
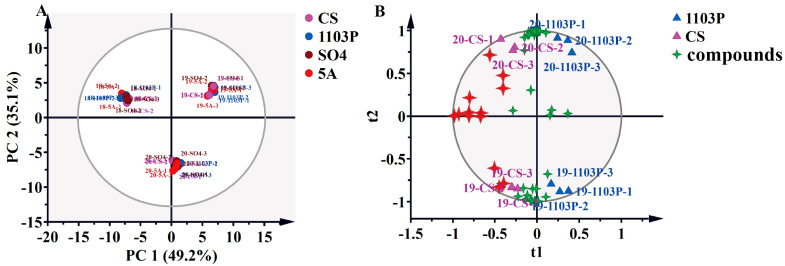
PCA based on aromatic substance concentrations of grapes in three vintages (2018–2020) (**A**); OPLS-DA based on aromatic substance concentrations of grapes in two vintages (2019–2020) (**B**). Note: The four-pointed red stars represent the differential compounds (VIP > 1), the specific names of which are shown in Appendix A. The results of the 200 cross-validation results are shown in Appendix A.

**Figure 5 foods-14-02695-f005:**
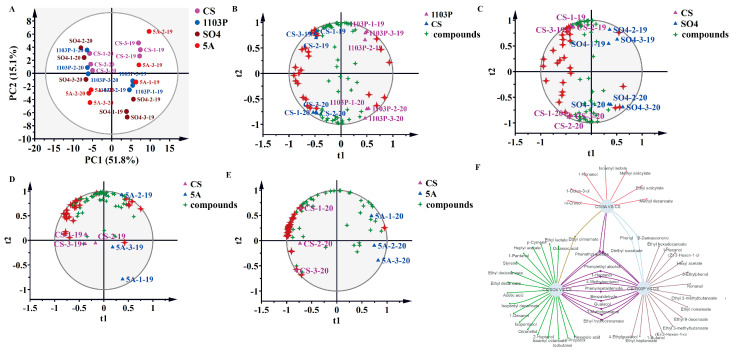
PCA based on aromatic substance concentrations of wine in two vintages (2019–2020) (**A**); OPLS-DA based on aromatic substance concentrations of wine in two vintages (2019–2020) (**B**–**E**); network diagram based on differential compounds (VIP > 1) in three rootstock combination wines (**F**). Note: The four-pointed red stars represent the differential compounds (VIP > 1), the specific names of which are shown in Appendix A. The results of the 200 cross-validation results are shown in Appendix A.

**Figure 6 foods-14-02695-f006:**
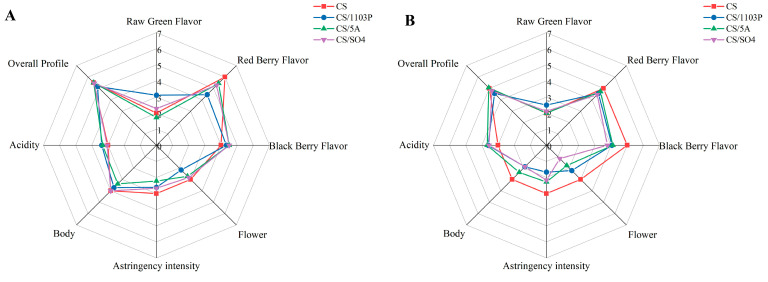
Radar map of wine flavor profiles for different rootstock combinations in 2019 (**A**). Radar map of wine flavor profiles for different rootstock combinations in 2020 (**B**).

**Figure 7 foods-14-02695-f007:**
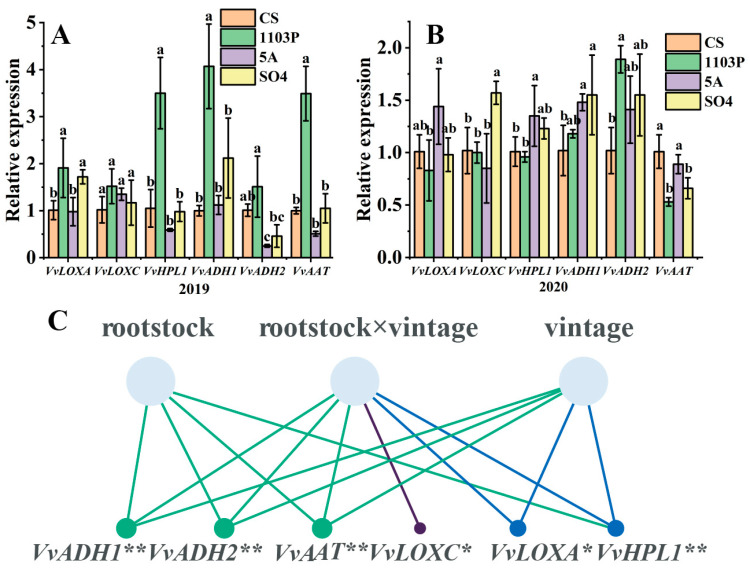
Gene relative expression levels of key genes responsible for volatiles in the LOX-HPL pathway in different grafted combinations of Cabernet Sauvignon grapes in 2019. (**A**) Gene relative expression levels of key genes responsible for volatiles in the LOX-HPL pathway in different grafted combinations of Cabernet Sauvignon grapes in 2020. (**B**) Two-way ANOVA of rootstock and vintage on key genes. (**C**) Note: different letters mean significant differences among the four treatments on the basis of Duncan’s multiple range test as *p* < 0.05. * indicates a significance level of 0.05, ** indicates a significance level of 0.01.

**Figure 8 foods-14-02695-f008:**
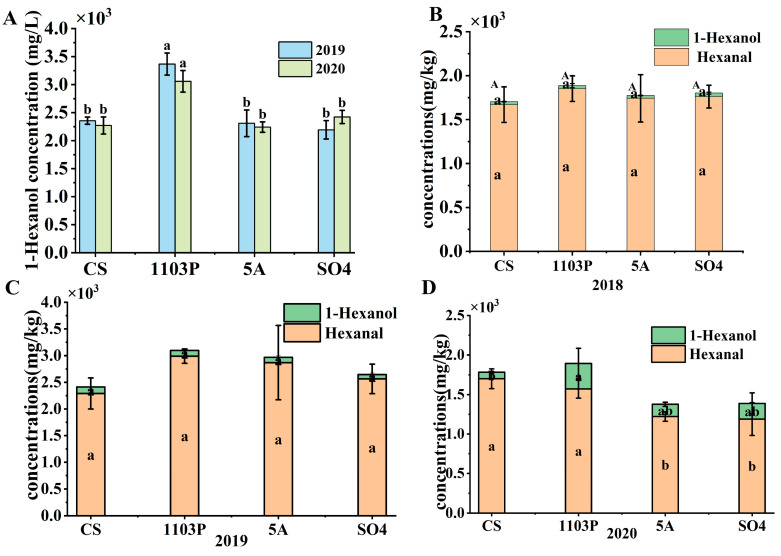
The concentration of 1-hexanol in wine in 2019 and 2020 (**A**); The concentration of 1-hexanol and hexanal in berries in three vintages (**B**–**D**). Note: different letters mean significant differences among the four treatments on the basis of Duncan’s multiple range test as *p* < 0.05.

## Data Availability

The original contributions presented in this study are included in the article/Appendix A. Further inquiries can be directed to the corresponding author.

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
