# Peer review of "High-Vigor Rootstock Exacerbates Herbaceous Notes in *Vitis vinifera* L. cv. Cabernet Sauvignon Berries and Wines Under Humid Climates"

_foods, 2025, doi:10.3390/foods14152695_

Round 1
Reviewer 1 Report
Comments and Suggestions for Authors
The title is descriptive but quite long and complex. A shorter, more focused title that reflects the main finding should be suggested.
The meaning of some abbreviations in the abstract is not clearly stated. This makes it difficult for readers to understand the study. Abbreviations that are unclear and do not appeal to the general reader should be avoided.
Some claims in the introduction section are presented without references. These points should be reviewed and supported by relevant literature.
The validity of the model in the PCA and OPLS-DA analyses is insufficiently presented. Criteria such as sample size, model overfitting risk, and cross-validation results should be reported more clearly.
The discussion section sometimes repeats the results section. These sections should be simplified and focus more on biological mechanisms and practical applications.
Why some rootstocks increase GLV levels while others do not should be explained in more detail. Stronger comparisons with the existing literature are expected on this subject.
The practical outcomes of the study should be clarified. Which rootstocks are not recommended for humid climates? What practical recommendations can be made for viticulture?
The limitations of the study have not been discussed. Points such as the dominance of the year effect, the limited sample size, and the focus on volatile compounds should be clearly stated.
There are numerous grammatical errors and expression issues throughout the article. Therefore, the study should be reviewed by a professional academic translation or language editing service.
The use of tenses is inconsistent throughout the text. Methods and findings should be written in the past tense, while general information should be written in the present tense.
The most important practical conclusions should be clearly and concisely highlighted. For example, 1103P increases the risk of green aroma and should therefore not be preferred in humid climates.
Tasting analyses require ethical committee approval as they are based on human subjects. It is not specified whether this approval was obtained.
The GC-MS data used for aroma analysis are only presented in graphs. However, chromatograms, retention times, and especially Retention Index (RI) values should be provided in tabular form.
What is the similarity percentage in the NIST11 library used to identify aroma components? The maximum and minimum similarity values should be clearly stated.
Was only MS used in the quantitative calculation of GC-MS peak areas? If verification with GC-FID was not performed, the reason should be justified. It should be clarified whether the quantities are directly based on GC-MS peak areas.
In Figure 6, aroma components are classified according to aromatic types. However, the contribution of these aromas to quality, which compound affects quality to what extent, has not been evaluated in terms of the literature.
The aroma quantities obtained in the study should be compared in depth with the data in the literature on the same type, and similarities/differences should be discussed.
Comments on the Quality of English Language
There are numerous grammatical errors and expression issues throughout the article. Therefore, the study should be reviewed by a professional academic translation or language editing service.
The use of tenses is inconsistent throughout the text. Methods and findings should be written in the past tense, while general information should be written in the present tense.
Author Response
Thank you very much for taking the time to review this manuscript. Please find the detailed responses below and the corresponding corrections in the re-submitted files.
1. Q: The title is descriptive but quite long and complex. A shorter, more focused title that reflects the main finding should be suggested.
A: Thank you for your suggestion. We have revised the title.
2. Q: The meaning of some abbreviations in the abstract is not clearly stated. This makes it difficult for readers to understand the study. Abbreviations that are unclear and do not appeal to the general reader should be avoided.
A: Thank you for your suggestion. We have changed the abbreviations in the abstracts, such as green leaf volatiles (GLVs), (Orthogonal partial least squares regression analysis) OPLS-DA
3. Q: Some claims in the introduction section are presented without references. These points should be reviewed and supported by relevant literature.
A: We appreciate your suggestion. We have shaded the changes in yellow and added appropriate references to some claims in the introduction.
4. Q: The validity of the model in the PCA and OPLS-DA analyses is insufficiently presented. Criteria such as sample size, model overfitting risk, and cross-validation results should be reported more clearly.
A: Thank you for your suggestion. We have added cross-validation results in the supplementary figure.
5. Q: The discussion section sometimes repeats the results section. These sections should be simplified and focus more on biological mechanisms and practical applications.
A: Thanks for your suggestion. We have censored the results of some of the replicated experiments and added the discussion.
6. Q: Why some rootstocks increase GLV levels while others do not should be explained in more detail. Stronger comparisons with the existing literature are expected on this subject.
A: Thank you for your suggestion. We have added a discussion on this issue and shaded it in yellow.
7.Q: The practical outcomes of the study should be clarified. Which rootstocks are not recommended for humid climates? What practical recommendations can be made for viticulture?
A: We appreciate your suggestion. Regarding the use of rootstocks, we give some recommendations in our conclusion.
8. Q: The limitations of the study have not been discussed. Points such as the dominance of the year effect, the limited sample size, and the focus on volatile compounds should be clearly stated.
A: We added to the discussion the limitations of the study, such as the ones you mentioned: the dominance of the year effect, the limited sample size, and the focus on volatile compounds, and the specific modifications have been shaded in yellow.
9.Q: There are numerous grammatical errors and expression issues throughout the article. Therefore, the study should be reviewed by a professional academic translation or language editing service.The use of tenses is inconsistent throughout the text. Methods and findings should be written in the past tense, while general information should be written in the present tense.
A: Thanks for your suggestions. We've reworked the grammar and tenses in the manuscript.
10. The most important practical conclusions should be clearly and concisely highlighted. For example, 1103P increases the risk of green aroma and should therefore not be preferred in humid climates.
A: Thanks for your suggestions. We have streamlined the conclusions in the manuscript
11.Q: Tasting analyses require ethical committee approval as they are based on human subjects. It is not specified whether this approval was obtained.
A: Thanks for your suggestions. We added ethical committee approval in 2.5
12.Q: The GC-MS data used for aroma analysis are only presented in graphs. However, chromatograms, retention times, and especially Retention Index (RI) values should be provided in tabular form.
A: Thanks for your suggestions. We have added retention time and retention index in the supplyment table S6 and S7, and we have added chromatograms in supplementary figures 1 and 2.
13. Q: What is the similarity percentage in the NIST11 library used to identify aroma components? The maximum and minimum similarity values should be clearly stated.
A: We added similarity percentage in the NIST11 library in Materials and Methods.
14. Was only MS used in the quantitative calculation of GC-MS peak areas? If verification with GC-FID was not performed, the reason should be justified. It should be clarified whether the quantities are directly based on GC-MS peak areas.
A: We only used MS for quantification, which has been labeled in the manscript. Based on many previous published literature, quantification using MS is feasible, but your valuable suggestion will be considered for use in our future experiments.
15. In Figure 6, aroma components are classified according to aromatic types. However, the contribution of these aromas to quality, which compound affects quality to what extent, has not been evaluated in terms of the literature.
A: Thanks for your suggestions. We added to the discussion shaded it in yellow.
16.Q:The aroma quantities obtained in the study should be compared in depth with the data in the literature on the same type, and similarities/differences should be discussed.
A: Thanks for the suggestion. We've added part of the discussion and shaded it in yellow.

Reviewer 2 Report
Comments and Suggestions for Authors
The manuscript "Modifications of the aromatic volatiles, sensory profiles and gene-expression properties of Vitis vinifera L. cv. Cabernet Sauvignon wine and grape by the rootstock × vintage effect" is well written, the idea of this paper is very interesting.
I am impressed by the approach and statistical methods used by the authors to analyze the obtained results. This requires a lot of effort and time.
However, despite my impression, I have a few remarks on how the quality of the paper could be improved:
Please provide information was the removal of grape stems performed prior to fermentation.
How long did the alcoholic fermentation take?
Malolactic fermentation was carried out, but how long?
Was the wine aged (and for how long), or was it bottled immediately after the completion of alcoholic fermentation?
Since these are red wines, the information about the total extract is very important for quality of wine, but I cannot find this data in the wine analyses.
Author Response
Thank you very much for taking the time to review this manuscript. Please find the detailed responses below and the corresponding corrections in the resubmitted files.
1. Q: Please provide information was the removal of grape stems performed prior to fermentation.
A: Thank you for your suggestion. We have revised the content and highlighted it in red.
2. Q: How long did the alcoholic fermentation take? Malolactic fermentation was carried out, but how long?
A: Thanks for the question; the alcoholic fermentation went on for about 7 days and the malolactic fermentation went on for about 1 month. We have revised the content and highlighted it in red.
3. Q: Was the wine aged (and for how long), or was it bottled immediately after the completion of alcoholic fermentation?
A: The wines were aged for approximately 1 month before being subjected to a sensory tasting.
4. Q: Since these are red wines, the information about the total extract is very important for quality of wine, but I cannot find this data in the wine analyses.
A: Thank you for your suggestion. We would love to complete the data on the dry leachate of the wines, but unfortunately we didn't have enough samples of the wines anymore now. We will value the valuable comments you provide us and complete this part of the data in the future experiments.
Round 2
Reviewer 1 Report
Comments and Suggestions for Authors
Thanks for making such thorough revisions to the manuscript. I've reviewed your detailed responses and the updated files, and I'm very pleased with the improvements.
You've successfully addressed all my feedback, from tightening the title and clarifying abbreviations to strengthening the introduction with references and enhancing the PCA/OPLS-DA validity. The discussion section is now more focused, and the added insights on GLV levels are excellent. I also appreciate the clearer practical recommendations and the inclusion of study limitations. The grammar and tense corrections are evident and significantly improve readability.
The additional details for GC-MS data, including retention times, indices, and chromatograms, are very helpful, as is the clarification on NIST11 library similarity. Your justification for MS-only quantification is noted, and the expanded discussion on aroma contributions and literature comparisons further strengthens the analysis.
Overall, these revisions have greatly enhanced the quality and clarity of your manuscript.
Author Response
Thank you for recognizing our revised manuscript because of your suggestions. Your suggestions have led to a very significant improvement in the quality of our manuscript.